# Parenting Challenges and Opportunities among Families Living in Poverty

Lana O. Beasley [1,*], Jens E. Jespersen [1], Amanda S. Morris [2], Aisha Farra [3] and Jennifer Hays-Grudo [4]

1   Department of Human Development and Family Science, Oklahoma State University,
    Stillwater, OK 74078, USA; jens.jespersen@okstate.edu
2   Department of Psychology, Oklahoma State University, Tulsa, OK 74106, USA; amanda.morris@okstate.edu
3   Department of Counseling and School Psychology, University of Massachusetts Boston,
    Boston, MA 02125, USA; aisha.farra001@umb.edu
4   Center for Health Sciences, Oklahoma State University, Tulsa, OK 74106, USA;
    jennifer.hays.grudo@okstate.edu
*   Correspondence: lana.beasley@okstate.edu

**Abstract:** Poverty-related stressors have been found to impact parenting behaviors which can result in adverse outcomes for children. The current qualitative study focused on understanding the challenges of caregivers (N = 70) living in poverty. The sample was diverse and included mothers, fathers, and grandparents raising grandchildren. Stories of caregivers were gathered to improve the understanding of families living in poverty in an effort to work towards changing how our world supports families that are vulnerable. Results indicate that families experiencing poverty and related risk factors experience challenges in the realm of child safety, education, and racism/prejudice. Families also discussed ways to improve their environment which included increased financial resources, increased access to high-quality healthcare and childcare, and positive environmental change. Note that the current study outlines the complexity of parenting in poverty and that associated challenges are intertwined. Recommendations are made to address systemic barriers at the individual and community level in an effort to better support caregivers experiencing adversity and parenting in the 21st century.

**Keywords:** parenting; poverty; adversity; resiliency; family support

## 1. Introduction

In thinking about parenting in the 21st century, many challenges come to mind. Some of these challenges include concepts of individual experiences and how this relates to recent parenting research as well as the impact of the current global coronavirus pandemic. Other ideas relate to the difficulties and opportunities of technology, changing laws and trends around cannabis and other substance use, discipline versus punishment, and other "hot topics" discussed in the popular press regarding parenting. One of the most pressing issues facing millions of families in the United States and around the world in the 21st century is the struggle for families to make ends meet. Families strive to keep their children safe, provide food daily, and deal with stigma and discrimination that reduce their ability to obtain resources for their children. In the United States today, roughly 1 in 5 children live in poverty using the 1960s Federal Poverty Level standard, and 2 in 5 children live in households considered low income, which is twice that level and more comparable to international poverty cutoffs (Schickedanz et al. 2021). Poverty is associated with additional risk factors for children's development. For example, many grandparents are currently raising grandchildren (Hayslip et al. 2015), often due to parents' difficulties with substance abuse, as the opioid epidemic and rates of parental substance use have increased during the pandemic (Czeisler et al. 2020). Immigrant families in the US and around the world face unique struggles and discrimination, and the Black Lives Matter movement and recent

events have brought to light many of the systemic disparities that exist for families of color. Challenges related to substance use and discrimination can deepen poverty or be a precipitate to poverty, reminding us that there is an intersectionality to the myriad of environmental stressors that families face. The current manuscript focuses on findings from the stories of families struggling with these and other challenges, providing evidence from the voices of parents experiencing different types of challenges related to parenting without sufficient basic resources.

Although the 21st century has brought unique and universal parenting challenges, it is important to understand the specific challenges encountered by families raising children living in poverty. These risk factors may include economic stress, substance abuse, neighborhood and family violence, mental health issues, lack of access to appropriate educational programs, children experiencing developmental delays, struggles with health and nutrition, and family literacy issues (Lander et al. 2013; World Childhood Foundation 2020). However, the underlying factor for many of these families is poverty. Although the definition of poverty has been reworked and modified throughout the years, poverty can be defined as "a state or condition in which a person, family, or community lacks the financial resources and essentials for minimum standard of living" (Chen 2022). Poverty creates major parenting challenges by impacting the way parents fulfill their parental role, making it difficult to give full attention to parenting duties (Ahmed and Kingsolver 2005; Banovcinova et al. 2014; Gupta et al. 2007). Economic deprivation can lead to parental depression and stress which directly impacts child well-being (Ahmed and Kingsolver 2005; Banovcinova et al. 2014). Families experiencing poverty are at a higher risk for child maltreatment (Kim and Drake 2018). Further, growing up in poverty directly affects biological systems and can negatively influence important foundations for later mental health (Pryor et al. 2019).

### 1.1. The Continuing Spread of Poverty

Families living in poverty face mounting and unrelenting pressures and barriers in everyday living that can inhibit personal and family growth and functioning, broadly including inadequate housing, poor community infrastructure, dangerous neighborhoods, limited options for purchasing food and other goods, substance abuse, absence of health insurance, poor educational resources and opportunities, longer commuting distances, and limited opportunities for reliable childcare (Edin and Kissane 2010; McLoyd 2021; Staveteig and Wigton 2000; Vernon-Feagans et al. 2012). Such conditions have been termed chaos indicators, as families who have to frequently make adaptations to meet the basic demands of daily life and family functioning tend to live in more chaotic environments (Evans and Wachs 2010; Roy et al. 2004). These chaos indicators can be crippling for individuals and families, and unfortunately, research suggests that those experiencing the poorest conditions are only falling further behind. For example, Evans and Wachs (Evans and Wachs 2010) found that chaos indicators have been increasing in prevalence among low-income families while remaining relatively stable among middle- and high-income families. Moreover, it has been documented that the concentration of child poverty has increased dramatically in both urban and rural regions of the United States due in part to an outmigration of young upwardly mobile adults from these areas, contributing to a less educated workforce and increased community risk factors (Corbett and Forsey 2017; Johnson and OHare 2004; Mokrova et al. 2017). Additionally, the current state of health and economic disparities in the United States has become more evident during the coronavirus pandemic as reported in a number of studies showing populations with more diverse demographics, lower education and income levels, and higher disability rates experienced hardship and adverse outcomes at disproportionately high levels (Abedi et al. 2021; Khatana and Groeneveld 2020; Lopez et al. 2021).

### 1.2. Unique Parenting Challenges for Families Living in Poverty

Developmental scholars have suggested that the proximal processes between parents and young children are often in jeopardy for families living under increasingly stressful and chaotic circumstances (Bronfenbrenner and Evans 2000; Conger et al. 2010). Moreover, researchers have posited that poverty-induced disruptions in family functioning could contribute to a number of parenting challenges associated with adverse outcomes for children. Along these lines, studies have shown that economic pressures first tend to affect the marital or partnering relationships and emotional states of caregivers, followed by a diffusion of these pressures into the child's caretaking environment (Mederer 1999; Wray 2015). The penetrating effects of poverty have notable adverse effects on parenting and child outcomes in terms of parental sensitivity and responsiveness. For example, it has been found that poverty-related stressors can promote insecure parent–child attachment relationships and harsher parenting conditions (Conger et al. 2010), contributing to poorer social emotional support and functioning in children (Wray 2015). Moreover, studies have indicated that parents who have high levels of chaos in their lives tend to possess lower levels of parental sensitivity and reduced motivation to take an active role in engaging with their children (Corapci and Wachs 2002; Johnson et al. 2008), as well as increased parental verbal interference, more ignoring of children's communication efforts, and a lower likelihood of giving children objects to play with or explore (Coldwell et al. 2006).

Global research initiatives examining effective parenting behavior have clearly illustrated the substantial advantage of employing balanced, authoritative parenting styles characterized by parental responsiveness, sensitivity, warmth, and communication (Baumrind 1991; Conger and Conger 2002). Poverty, however, has been found to play a significant role in influencing how parents interact with and raise their children (Conger and Conger 2002). As such, it has been found that parents living in poverty are less likely to use nurturing parenting practices and more likely to use authoritarian or inconsistent parenting styles characterized by harsh interactions, including the use of corporal punishment (Wray 2015). Research has shown the deleterious effects of children living in poverty coupled with harsh parenting styles, including exacerbated child behavior problems, such as elevated externalizing behaviors across childhood (Coldwell et al. 2006; Hao and Matsueda 2006; Scaramella et al. 2008). In a study examining the consequences of poverty on parenting and early childhood, Scaramella and colleagues (Scaramella et al. 2008) found harsh parenting and child externalizing problems to be present across three generations, further illustrating the long-lasting effects associated with growing up in poverty. Furthermore, children raised in poverty are more likely to experience early trauma and neglect at the hand of problematic parenting and family relations (Steele et al. 2016). As such, children who grow up in poverty not only carry the burdens of greater health and educational problems but these problems can be perpetuated by maladaptive parenting styles and behaviors (Wray 2015). With the myriad of potential negative impacts of poverty on the family system, it is vital to understand parenting in poverty. Due to the rapidly growing diversity in family make-ups, the current study focused on understanding parenting among a diverse sample of caregivers including mothers, fathers, teenage parents, mothers in recovery from substance use, grandparents raising grandchildren, and Latinx mothers and fathers. Based on the current state of the literature we targeted these family types due to their underrepresentation in research related to poverty (Edin and Kissane 2010; McLoyd 2021). Research associated with parenting issues in different types of families types is outlined in the following section.

### 1.3. Some Characteristics of Parents Living in Poverty

Adolescent parents. While teen pregnancy rates have been declining in the United States over the past fifty years, rates have remained notably higher among lower-income groups, including minority populations (Hamilton et al. 2015). Furthermore, the highest rates of teen pregnancy occur in states with larger populations of racial/ethnic groups and wider margins of income inequality (Ventura et al. 2014). Due to the elevated social, health, and economic risks associated with teenage parenting, it has received much attention

in terms of research and social and public health policy development (Smithbattle 2018). Despite increased advocacy, teen parents still face enormous challenges that can be difficult to mitigate, particularly among teens living in poverty. For example, low-income teen moms are less likely to gain access to family planning resources (Smithbattle 2018), have limited educational attainment (Bell et al. 2014), and report reduced hope about the future (Fedorowicz et al. 2014).

Parents who are recent immigrants. Along with the traditional challenges that come with raising children, parents who were born abroad, or for whom English is not their language of origin, have been found to experience increased parenting stress, lower self-efficacy, and less adaptive parenting strategies (Abraham et al. 2018). For example, Nam and colleagues (Nam et al. 2015) found that nativity status accounted for 63% of the difference in parenting stress when comparing Latinx and Caucasian mothers. Coupled with higher parenting stress and lower parenting confidence, it has also been documented that recent-immigrant families possess elevated challenges in securing psychological and physical health resources to support their parenting efforts, while also experiencing an adjustment period prior to securing employment upon arrival (Armstrong et al. 2005; Conger et al. 2010; Duncan and Trejo 2012). Moreover, similar parenting studies have shown that linguistic and cultural barriers, including perceived discrimination, can obstruct parents in gaining access to beneficial programs and opportunities (George et al. 2014).

Mothers recovering from substance abuse. In 2019 there were approximately 1.4 million adults who entered substance abuse treatment in the United States (Substance Abuse and Mental Health Services Administration 2019). Studies have suggested that up to 70 percent of women entering substance abuse treatment programs were parents of children under the age of 18 (Greenfield 2002; Rubenstein and Stover 2016). While recovery efforts are encouraging, it has been found that mothers recovering from substance abuse experience unique risk factors including enhanced vulnerability to physiological consequences, increased prevalence of mental health problems, poor nutrition, increased rates of relationship problems, and lower reports of social support (Greenfield 2002; Hernandez-Avila et al. 2004; Niccols et al. 2012). Moreover, addiction treatment and recovery do not erase the long-term effects that substance-induced behavior, domestic violence, poverty, and poor parenting can have on children (Murphy and Ting 2010).

Single parents. Recent data indicates that single-parent households are on the rise in the United States (Coles 2015). While in the past individuals often acted as single caregivers due to divorce or widowhood, recent trends in cohabitation and non-marital births have contributed to non-marital parenthood accounting for over one-third of single households (Coles 2015; Wildsmith et al. 2011). Given the accelerated rates with which single individuals are primary caregivers, increased attention has been given to single-parenting behavior. In many cases, single parents have been found to be adaptive to their single-parenting environment. For example, studies have shown that given the unique expectations and demands of single parenting, fathering behavior tends to override traditional gender differences (e.g., in household work, parental involvement) resulting in single fathers displaying parenting behavior that is more similar than different to that of single mothers (Coles 2015; Hook and Chalasani 2008). Despite these similarities, however, it has been found relatively consistently that single fathers are less likely than single mothers to provide supervision, monitoring, and closeness and intimacy with their children (Bronte-Tinkew et al. 2010; Demuth and Brown 2004).

Grandparents raising grandchildren. Research has shown that grandparents who raise their grandchildren are a positive influence on their grandchildren, particularly among children being raised in poverty (Hayslip et al. 2019). Despite the important role that these grandparents play in the lives of their grandchildren and the associated fulfillment of supporting their grandchild(ren), it has been found that grandparents report feeling removed from their same-age peers, experience increased physical and emotional challenges associated with parenting, experience shame associated with the stigma of raising one's grandchild, and may feel judgment from others regarding their own child's

failures as parents (Hayslip et al. 2019). Moreover, grandchildren of custodial grandparents have been found to possess deficits in emotional, social, and behavioral development compared to normative samples (Harnett et al. 2014). Regardless of these findings, children being raised by their grandparents have been found to function more adaptively than children raised in nonrelative foster care (Hayslip et al. 2019). Finally, despite custodial grandparenting cutting across gender, racial, and ethnic lines, low-income individuals have been found to be disproportionately represented (Fuller-Thomson et al. 1997; McDaniel and Connidis 1990).

Despite the variations in circumstances, needs, and challenges among these family types, poverty is a common set of circumstances that exacerbates the difficulties that families of diverse backgrounds experience. By seeking to understand the perspectives of these families and both their common and unique challenges, we can gain a better perspective for how and where intervention efforts can begin and where they might continue.

### 1.4. Current Study

It is clear that poverty is a risk factor that confers additional risk among families already experiencing economic and other challenges. It is also evident that to mitigate this risk, we must enhance our understanding of the parenting challenges families face to identify supports that promote resilience. To address these goals, the current qualitative study aimed to understand the challenges a variety of families living in poverty face (i.e., mothers, fathers, teenage parents, mothers in recovery from substance abuse, grandparents raising grandchildren, and Latinx mothers and fathers) when caring for their young children. While numerous studies have examined associations between poverty and individual factors of family life or child development (Banovcinova et al. 2014; Edin and Kissane 2010; Gupta et al. 2007; McLoyd 2021; Pryor et al. 2019), the current study was designed to provide a more comprehensive narrative of the experiences that a variety of parents living in poverty experience on a day-to-day level, with the intent to inform community intervention and policy development for families living with increased risk factors while parenting young children. While qualitative methods have not been traditionally utilized for conducting research related to parenting and poverty at large, qualitative research examining such themes has proven influential and uniquely informative of the realities of parenting in poverty (Edin and Kissane 2010; Schickedanz et al. 2021). The strength of the present study is the ability to capture parents' own experiences related to parenting and raising their families in poverty at the micro-level. Moreover, this study uniquely presents the perspectives and experiences gathered from a diverse sample of parents, including mothers, fathers, grandparents, single parents, parents recovering from substance abuse, and parents from diverse racial and ethnic backgrounds (see Table 1).

**Table 1.** Participant Demographics (N = 70).

| Sample Characteristics | % |
| --- | --- |
| Caregivers | |
| Mothers | 54.3 |
| Fathers | 28.6 |
| Grandparents | 17.1 |
| Number of Children | |
| 1 | 29.3 |
| 2 | 30.7 |
| 3 | 21.3 |
| Education | |
| 11th Grade or Less | 25.4 |
| HS Diploma/GED | 25.3 |
| Some College | 26.7 |
| Two-Year Degree | 8.0 |
| Four-Year Degree | 9.3 |
| Graduate Training | 2.7 |

**Table 1.** *Cont.*

| Sample Characteristics | % |
|---|---|
| Race | |
|     Hispanic/Latino | 31 |
|     European American | 27 |
|     African American | 19 |
|     Native American | 10 |
|     Asian American | 6 |
|     Multi-Racial | 7 |
| Adolescent Mothers | 10 |
| Recovering from Substance Use | 9 |

Note. HS: High School; GED: General Education Development.

## 2. Materials and Methods

### 2.1. Participants

A purposive sampling method was used with participants that are caregivers of young children (N = 70). Caregivers included mothers (N = 38), fathers (N = 20), and grandparents raising grandchildren (N = 12) with data collected in the fall of 2016. Poverty status was based on eligibility for state-assisted programs (i.e., WIC-eligible, Head Start eligibility, etc.). Approximately half of the sample (44% of participants) indicated an annual income of USD 20,000 or less with the majority of families having one (29.3%), two (30.7%), or three (21.3%) children. In terms of education completion, 25.4% of participants reported completing 11th grade or lower, 25.3% graduated from high school or had a GED, 26.7% reported some college or technical skills, 8% were graduates of a two-year college program, 9.3% graduated from a four-year college, and 2.7% reported postgraduate work. All participants were invited to participate in the study by phone and given a brief description of the topic of the discussion. Recruitment was conducted by a local marketing firm or research team members.

Overall, the sample was diverse with 31% of caregivers identifying as Hispanic/Latino; 27% as European American; 19% as African American; 10% as Native American; 6% as Asian American/Pacific Islander; 7% as multi-ethnic. Forty-three percent were married and over half of the sample was employed (67%). It is also important to note that 9% of the sample consisted of caregivers recovering from substance use and 10% were adolescent mothers. All demographic information is presented in Table 1.

### 2.2. Procedure

A total of seven focus groups were conducted with each group ranging in length from approximately 1 to 1.5 h each. Focus groups were conducted by a trained qualitative moderator with all groups including a trained notetaker so that additional discussion could be encouraged if the moderator missed an important area to probe. Interviews were transcribed by a leading transcription company and all transcripts were cross-checked for accuracy by research members. Focus groups with Latinx mothers and fathers were conducted in Spanish with all interviews translated and transcribed by a Spanish-speaking research technician. Additionally, coding for Spanish-speaking transcripts was conducted by a group of Spanish-speaking qualitative team members that also conducted the interviews, to ensure cultural content was not lost in translation. A total of 70 participants were in focus groups with 6–12 participants per group; all data were collected in a mid-sized, southern Midwest city during July and August of 2016.

Focus group participants received a USD 100 gift card to compensate them for their time and a USD 25 gas card to assist with travel expenses to the group. All focus groups were conducted in the evening with a small meal provided for participants. All study data collection and evaluation methods were approved by the University Institutional Review Board (IRB).

Focus group interview guide. An interview guide for focus groups was developed by the lead qualitative researcher and reviewed by the qualitative research team. The interview guide was then reviewed and revised with feedback from collaborative partners. Questions were based on an understanding of previous research that families in poverty can face numerous challenges. Therefore, the purpose of the semi-structured guide was to assess parenting challenges experienced by families living in poverty. Specifically, caregivers were asked about "problems parents face", "things that would make life easier for you and your family", and "immediate needs of your family". Other questions were focused on understanding services that families are aware of within their community and their experience with services. Time was spent with participants explaining the purpose of the study, confidentiality, and their choice to answer or not answer questions, depending on their comfort level. At the beginning of the interview, participants were thanked for their attendance in the focus group and how much the research team appreciated their expert opinions. Participants were also given a general demographic survey prior to participating in the focus groups. It is important to note that the topic of caregiving challenges in today's world emerged as a theme throughout the interviews with all data included in the current manuscript.

### 2.3. Data Analysis

Qualitative data analysis of the transcriptions was conducted using NVivo 10 software. A template approach (Patton 2002) was used to identify broad themes from the focus group data. This approach involved developing a coding "template" that included hierarchical coding of broad themes, using codes developed before examining the data, and developing an initial template that is applied to the larger set of qualitative data. This iterative process included using content analysis to identify core constructs and themes emerging from the data. More specific themes were also identified and coded as sub-codes within the broader categories using the aforementioned template, or codebook, developed by three to five trained qualitative researchers. Throughout, analysis discrepancies were discussed to ensure the reliability of coding. This approach was used to gain an understanding of the perceptions of participants regarding parenting challenges they have faced. All focus groups were transcribed and cross-checked during analysis to verify accuracy. Importantly, initial themes were explored for different focus groups and categories of caregivers, and because similar themes emerged across groups all data were combined for coding and analysis. Themes are described below with representative quotes to illustrate key findings.

### 3. Results

### 3.1. Problems Parents Face

Child Safety Concerns. Caregivers indicated, in high frequency, types of child safety concerns. A specific concern was issues with living in "bad neighborhoods" that led to situations that created safety concerns for families with one caregiver explaining that "it's pretty routine that we have people—like a SWAT team on our street". Other themes that help illuminate child safety concerns in neighborhoods include fear of outside play, risk of child kidnapping, drive-by shootings, cars speeding through neighborhoods. Regarding drive-by shootings, a caregiver shared, "that's really a big issue . . . because it's a lot of stuff going on at the moment, to where, you know, like, our kids ain't able to go outside and play without somebody drive-by shooting". Another caregiver explained "and I don't know if the kid died, but he got shot, that's the point. And so I told him (my child), you know, that's why I don't like them playing outside . . . " Other caregiver concerns surrounding child safety include distrust of police, not trusting others with their children, and exposure to negative influences. Regarding difficulty trusting others to care for children a family explained, "a daycare worker . . . left a kid in the van, after they went on a field trip". Although it was reported that the child was not severely injured in this situation, the parent explained the fear that is experienced when you realize you cannot trust childcare providers.

Education and Care. Many caregivers reported that education is a problem families face. In terms of childcare and early childhood education, caregivers reported issues in finding affordable, high-quality childcare. Specifically, they discussed issues with finding care when school is out (after hours and summer) and finding childcare programs that have extended hours that fit with variable work schedules. Additionally, the prohibitive cost of high-quality childcare was discussed and the theme of difficulties in finding trustworthy providers emerged again.

In terms of childhood education, families indicated that due to budget cuts, the education system has suffered. Families also reported concerns with their children being bullied within the school setting and schools not being able to support children with disabilities. Large class sizes were discussed as an issue, resulting in children not getting the individual attention they need to learn. Caregivers also reported problems pertaining to reduced bus stops and school violence with one caregiver explaining, "there was a potential kidnapping right by my house—so their kids are stuck in the bus stop from like 7:30 until, I don't know . . . 8:30? The whole time waiting for the bus by themselves. Anything can happen". All of these issues culminated with caregivers explaining the inequality within the educational system and a lack of good educational options for their children. Families discussed this inequality results in negative influences at school and the desire to move to a "better" school district for their child(ren) with many limitations to reaching this goal. One caregiver explained:

> "We're thinking about moving him to another school, one where he can learn better, because since he's little and then because of the influences, because—the influences of the other kids also has a lot to do with it for him, because he even had some bad behavior sometimes, when he didn't have it before".

Racism and Prejudice. Racism can be defined as prejudice, discrimination, or antagonism directed against a person or people on the basis of their membership in a particular racial or ethnic group, whereas prejudice can be understood as a preconceived opinion that is not based on actual experience. Related to racism, parents reported that minorities face explicit and implicit racism within their communities. It was further explained that there is a lack of cultural sensitivity with some families discussing issues of immigration status negatively impacting families. One family shared the trauma of having their children's father deported and another discussed the negative impact immigration status can have on child educational opportunities. An example quotation in this theme includes:

> "To my dismay, a few months ago he was deported . . . we fought and argued but where girls (our children) do not listen . . . now that he was deported . . . it was not divorce what caused the problem of my child . . . and that hurts my heart because you imagine a nine-year-old girl crying in the corners and suffering for their dad, this, sent her to a psychologist, the medicated".

Other areas of prejudice included fathers reporting issues with child custody and difficulties within the court system in obtaining rights to their child(ren). Fathers further disclosed that some judicial systems favor mothers, regardless of their ability to care for the child(ren). This prejudice creates a myriad of issues including potential child safety concerns due to courts not understanding the strengths and weaknesses of each parent as well as decreased father exposure. Fathers also explained they have been stereotyped as less competent caregivers which can make it difficult for them to be respected by society in their parenting role. Additionally, it was explained that this stereotype is even more pronounced towards minority fathers with them being seen as less effective in their parenting role. One father explained, "being an African American, people already look at you and say, 'Hey, you're not gonna be there for your child'. It's like a stereotype, see what I mean".

A prejudice disclosed by grandparents involved issues surrounding their legal rights to their grandchildren. These issues were reflected as difficulties getting medical care and resources. An example quote of a grandparent having difficulty obtaining resources includes:

"So I went down there because I needed a car seat . . . they said, 'Well, are you the legal guardian of the child?' I said, 'No, but I have the child every day . . . I need a car seat, and I don't have any money right now, 'cause I work with school and so I don't get paid in the summertime' . . . they wouldn't give me the car seat".

Grandparents also disclosed a prejudice that they made poor decisions with their children and that is why they are now raising their grandchildren. One grandparent shared that "because we have a stigma that we're the reason our kids are the way—you know what I mean? I mean, realistically it's like we have a stigma".

Women recovering from substance use also discussed prejudice they face that impacts their ability to parent. Some of this prejudice was related to the repercussions of having a felony with a participant explaining:

"Like there's so many things—there's so many barriers to being where you need to be.

It just gets overwhelming. So part of what I hear you say is reducing some of the barriers

like, um, all the stigma surrounding the felony".

One specific repercussion participants shared was the numerous fines associated with a felony that make it difficult for parents to start a new life. It was also shared that "it's hard to actually find housing with having a felony . . . you're a felon, you've got these kids, but then you have no assistance to help you get any kind of housing". Another difficulty disclosed was obtaining employment, especially employment that pays higher than the minimum wage. It was explained that all of these issues make it problematic for women in recovery to provide for their families.

It is important to note that all families discussed issues related to prejudice they experienced in their life, often related to living in poverty and their life circumstances. Areas of racism were more strongly shared among families that indicated a minority status.

*3.2. What Would Make Life Easier?*

Given the aforementioned parenting challenges, it is important to understand what would make life easier for families. Families discussed the need for increased resources related to (a) finances, (b) healthcare/childcare, and (c) the environment.

Financial Resources. One area caregivers identified that would make life easier included having access to financial resources. This included obtaining better-paying jobs which led to a discussion surrounding the need for an increase in the federal minimum wage. Parents also discussed the need for increased financial literacy with a parent explaining:

"At the moment, since me and my husband are struggling to make it for our family, it's like, all I hear is, 'Your credit score is too low'. Yes, I know my credit score is too low. I'm working on it. But there's nothing too much that I can do about the past because my current debt is more important".

Related to financial resources, parents also indicated the desire to improve their education. The discussion included parents wanting access to GED classes, career training, and other educational opportunities.

Healthcare and Childcare. Other themes that emerged included better access to healthcare and high-quality childcare/daycare for children. Specific to childcare, families discussed the need for high-quality care for children with disabilities, before and after school care, and care when children are sick. One caregiver explained:

"I know they're not supposed to take care of our sick kids, but when you're working and you can't leave and you have one hour to go get your child, like sometimes you are not physically able to . . . " "Or you could lose your job". "You're gonna lose your job which means you're not gonna be able to support your family and . . . I don't know what that solution is, but that has been my obstacle recently".

Environment. Other areas that would improve family life include creating a safer environment for children (e.g., housing, school), affordable/accessible housing, increased access to transportation, and improving the family support system. The theme of increasing the family support system involved needing increased support in the home which includes help with household duties with families reporting wanting a "nanny" or a "housekeeper". It was also discussed that programs need to support families having quality time together with a caregiver sharing, "once a year . . . a program that could let a family go on vacation together, spend that quality time, because I think for a lot of us, our biggest issue is we're working jobs . . . we just don't have that time to really bond with our children".

## 4. Discussion

The primary goal of the current study was to improve the understanding of the challenges families living in poverty face in parenting in today's world. It is clear from the aforementioned themes that low-income families face a multitude of systemic problems that impact their ability to meet the demands of parenting.

Through connecting the stories that families experience, an underlying theme of the struggles associated with poverty emerged across families from diverse backgrounds and living arrangements. In fact, many families experienced a cycle of poverty that continues to worsen due to social as well as economic inequities. Exposure to violence and crime was intertwined with issues of budget cuts and the impact this had on education. This was perceived as causing a reduction of bus stops with families being fearful that an increase in crime could lead to violence and kidnapping as children must now travel farther or wait extended times to access bus transportation to school. This was also evident in themes emerging for lack of access to high-quality childcare and fear of children being harmed when high-quality care facilities are not available to families. Additionally, this emerged within the theme of not trusting others with their children within school and childcare settings due to reports of children being harmed in care. Despite these fears and concerns, parents are consistently faced with placing their children in unsafe environments and situations that pose a potential threat to their safety and well-being in order to work (Hanson and Carta 1995). While parents may have good intentions to protect their children by limiting their outside play or isolating them from others, this may have serious implications for their health, academic performance, and psychological well-being (Dowdell et al. 2011; Lacey et al. 2014).

Additionally, there was an interconnecting theme between education and racism/prejudice and the intersecting of marginalized identities. Caregivers noted their concerns about their children being bullied, and not being able to receive support for their children with disabilities or special needs. Bourke-Taylor and colleagues (Bourke-Taylor et al. 2010) interviewed eight mothers who had a child with a disability and found that working was not an option for many mothers. Further, it was found that childcare services that can support children with disabilities were non-existent. This is important to outline given that low-income parents are either working full-time or working multiple jobs to meet financial demands (Rispoli et al. 2018). This is especially important for childcare given that caregivers in the present study noted difficulties in finding childcare with extended hours that would fit with their work schedules. Parents experiencing multiple risk factors with children living with disabilities may be faced with untenable decisions when making the choice to either work to meet financial demands or take care of their child(ren). This statement reflects the comments stated in themes related to financial resources, with childcare as a possible solution for caregivers to better support their child(ren).

Many families were cognizant of the inequities within the educational system and the repercussions it may have for their children. One example provided is the large classroom sizes, limiting the individual attention that children need to learn. Studies have noted the benefits that accrue when low-income children who are racially and ethnically diverse are able to engage individually with their teachers (Ferguson et al. 2007; Hamre and Pianta 2005; Sabol et al. 2018). Unfortunately, large classroom sizes may impact the ability of children to

engage with their teachers. Ferguson et al. (2007) found that student engagement is lower for those who are located in low-SES neighborhoods and low-income families. Therefore, if a child does not receive adequate attention, and falls behind in their academics, it is difficult for them to catch up (Reardon 2011). Early childhood programs are the best way to impede these effects (Sabol et al. 2018), but many of our participants noted they are unable to access high-quality early childhood education for their child(ren).

Caregivers also experienced a wide range of racism and prejudice that impact how they and their children are perceived and treated. This was evident by hearing the stories of families who hold a different immigration status, who hold different ethnic/racial identities, and who have different family and caregiving dynamics. Hearing descriptions about the racism and prejudice they experienced demonstrated that even when systems are put in place, workers within the system can deter caregivers from obtaining the support they need to support their child(ren). This is seen in the earlier example, where a daughter was experiencing difficulties with her father being deported and was referred to a psychologist. While a psychologist can be beneficial in addressing mental health concerns, different cultures perceive mental health issues as taboo or hold a stigma against them (Hirai et al. 2021). In addition, cultural sensitivity is needed within practitioners to understand the varying experiences of different ethnically and racially diverse groups. Thus, this example demonstrates that even if accessibility and affordability of services are increased for these populations, service providers may be less than beneficial due to a lack of cultural sensitivity. This cultural sensitivity is also needed for grandparents who have different caregiving roles within their family dynamics and hope to obtain legal rights. Service providers need to understand and acknowledge that caregiving goes *beyond* the biological parents, and grandparents may play the same role (or more of a role) of caregiving within the family (Lent and Otto 2018; Pashos and McBurney 2008; Schneiders et al. 2021; Xu et al. 2017).

Additionally, being a father and caregiver also looks different for families living in poverty with associated risk factors. While it is important to acknowledge they face difficulties in the court system in obtaining rights to their children compared to the mothers, they also perceive being stereotyped as less competent caregivers. Thus, caregiving and parenting may be viewed differently depending on age, gender, and racial/ethnic background. These stereotypes and prejudices can impact the treatment caregivers receive from social services and educational and judicial systems.

One notable finding that is evident from these caregivers is that they are trying to parent the best way they can, but they are struggling to do so given the structural and systemic barriers that are in place. This is evident among caregivers who have experienced a felony and are trying to start a new life with their family but are unable to do so without government assistance. The system punishes them for their previous crimes; thus, impacting their ability to receive services, assistance, or housing. It was also evident among all caregivers interviewed that they are completely dedicated to raising healthy, intelligent children.

It is also important to note that in discussion related to what would make life easier for families, each of the themes that emerged could be tied back to specific challenges except for the theme of racism and prejudice. One hypothesis could be that racism and prejudice are so prevalent within the lives and experiences of families experiencing poverty that it might be viewed as an intangible construct, difficult to directly address. One could also argue that many of the themes that would make life easier could be related to systemic issues related to poverty, as well as racism and prejudice.

These results demonstrate that families living in poverty are combating not only the lack of material resources but a number of societal and systemic issues that are closely intertwined with poverty. Thus, while addressing one challenge may indeed be helpful to these families, it may not solve other related problems.

### 4.1. Limitations

The first limitation is that this study was conducted in an urban community and may not encompass either similar or unique barriers related to parenting in rural communities. A second limitation is a problem with the generalizability of the findings. The sample includes a diverse representation of caregivers living in poverty, but not all potential groups experiencing poverty were represented (e.g., incarcerated parents), and findings may not be generalizable to families living in poverty outside of the U.S. Indeed, many of the challenges identified by caregivers may be specific to a local context or local policies, and this should be taken into account when relating the findings to families living in poverty more generally.

Another potential limitation is the study was conducted prior to the COVID-19 pandemic. The COVID-19 pandemic has impacted the world in numerous ways and has created a myriad of parenting challenges. Enforced isolation, school closures, and hybrid learning have changed the home environment for many families and the way caregivers parent. COVID-19 has increased economic disparities, psychosocial stress among caregivers, alcohol and substance use among parents, and may have created new problems for families experiencing risk. This is particularly important to emphasize given how the COVID-19 pandemic has altered the way individuals physically engage with each other and experience isolation (de Figueiredo et al. 2021). Therefore, families experiencing adversity may need additional solutions to their challenges. Even so, the current study collected a broad range of stories from diverse families, which vastly enhances our understanding of the issues families face parenting in today's world. Additionally, the wide range of caregivers that we spoke to alluded to many similarities families experience, as well as unique and specific challenges.

### 4.2. Conclusions

The current study sought to understand the challenges experienced by diverse low-income families as they parent their young children in today's world. Our findings suggest that families living in poverty experience challenges in the areas of child safety, education, and racism/prejudice that are likely to limit their children's health and development. Increased financial resources and opportunities, improved access to healthcare, high-quality schools and childcare, and creating positive e changes within communities, neighborhoods, and home environments were suggested by parents as possible solutions to address their challenges. Given the intersection of many of the challenges experienced by parents living in poverty, policy and program changes that address individual, community, and system-level issues are most likely to be effective. In this respect, the challenges facing low-income parents in the 21st century are remarkably similar to those facing low-income and otherwise marginalized parents of the 20th century. As noted in 1976 by Dr. Ed Ziegler, the first director of the U.S. Office of Child Development and one of the designers of Head Start, "In our nation today children and families all too often come last, and the social barriers to providing a better quality of life for our nation's children have become almost insurmountable" (Morris et al. 2021). It is the hope of these authors, and the parents whose stories they heard, that the 21st century will see an increase in the value we place and the support we provide parents and children.

### 5. Implications for Parenting in the 21st Century

Parenting in the 21st century holds many challenges for families, and these challenges are potentially greatest among families living in poverty. The following recommendations were formed through the qualitative data gathered in this study in an effort to understand the problems families living in poverty face and what they perceive to be potential solutions. Many of these solutions have implications for the delivery of social services and programs for low-income families, as well as local and federal policies.

One recommendation is to review current policies regarding minimum wage. It is clear that families are not able to support their families with the current minimum wage

salary. This translates to families not having basic needs met (food, shelter, clothing, etc.) which leads to families being challenged in meeting the higher-level needs within the home and environment (attachment, nurturing, educational, etc.).

Another recommendation based on parents' reported concerns is the need to review policies that fund access to high-quality education and care for children and families experiencing poverty and other risk factors. This includes access to high-quality early childhood education programs and after-school programs, as well as access to schools that are able to provide a safe and high-quality level of education. Many parents expressed concern with not having good educational options and the inability to move their family to better school districts due to financial constraints. Therefore, efforts need to ensure that all schools can provide the education and accessible services that children need. In the community we studied, this access also requires improvement in public transportation systems for families. Many families discussed the effects of not having adequate transportation for accessing medical care, educational options (caretaker and child), employment options, and basic needs (groceries, resources, etc.).

Increasing the availability of safe, affordable housing is another recommendation that has a direct impact on children's health and development. Specifically, when children are not provided with safe environments they are not able to play and explore outdoors, preventing the acquisition of skills and independence. Additionally, the negative events that children in violent communities are exposed to relate to the cumulative toxic stress exposure which impacts their short-term and long-term development.

Our last recommendation is to work towards changing the climate of how we approach cultural diversity. Specifically, many families reported living in fear due to oppression of their race, ethnicity, age, gender, prior criminal history, and immigration status. This includes developing and training our judicial, educational, social services, and healthcare systems in trauma-informed practices so that caregivers are supported rather than ostracized or mistreated.

**Author Contributions:** Conceptualization, L.O.B. and A.S.M.; methodology, L.O.B.; formal analysis, L.O.B.; data curation, L.O.B. and A.F.; writing—original draft preparation, J.E.J., L.O.B. and A.F.; writing—review and editing, A.S.M. and J.H.-G.; funding acquisition, A.S.M. All authors have read and agreed to the published version of the manuscript.

**Funding:** This research was funded by the George Kaiser Family Foundation.

**Institutional Review Board Statement:** The study protocol was approved by the Institutional Review Board of Oklahoma State University. The IRB code is HE1651.

**Informed Consent Statement:** Informed consent was obtained from all subjects involved in this study.

**Acknowledgments:** We would like to acknowledge the hard work done by Amanda Morris' Child and Adolescent Development Lab, Consumer Logic, and Saxum. We are specifically grateful to Madison Pace from Saxum for her efforts in organizing the focus group data collection. We would like to thank qualitative team members: Jordan Love, Martha Roblyer, Amy Huffer, Amy Anderson, Sandy Boyaci, Jerry Root, Samuel Watson, Corie King, and Rebekah Joseph. Without such a strong collaboration and a team that is passionate and dedicated to helping children and families this project would not have been possible.

**Conflicts of Interest:** The authors declare no conflict of interest.

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
