# Peer review of "Parenting Challenges and Opportunities among Families Living in Poverty"

_socsci, doi:10.3390/socsci11030119_

Round 1

Reviewer 1 Report

The manuscript is a qualitative study that aims to understand the challenges a variety of families living in poverty face when caring for their young children. It is interesting to focus on the understanding of families living in poverty but I have some concerns on the manuscript:

  1. The Current Study section needs revision. It is important that autors mention the previous literature focus on this topic. For example, in the sentence “While numerous studies have examined associations between poverty and individual factors of family life or child Development…” it would be beneficial for the authors to explain which studies are and by which authors. Likewise, the authors continue “the current study was designed to provide a more comprehensive narrative of the experiences that parents living in poverty experience on a day-to-day level, with the intent to inform community intervention and policy development for families living with increased risk factors while parenting young children”, please, make sure all objectives and hypotheses are stated in this section and explain the results expected. Moreover, the paragraph corresponding to lines 53 to 63 that are in the Introduction section does not seem to fit the authors' argument. Perhaps, it would be more appropriate to include this paragraph in the Current Study section.
  2. Create a table would be useful in order to describe the characteristics of the sample.
  3. I believe that Measures section is poorly described and needs to be revision and amplified beyond to carry out an interview guide. To understand the study it is necessary more information about the interview carried out. Same with Data Analyses section…for example, What codes was each themes coded with?...
  4. Authors could improve their literature review and make an effort to connect their study y what are the uniqueness of parenting features and parenting challenges that parents face in the 21st century as they have done in the Introduction section.
  5. Please, revise the limitations of the study, maybe one of these could be that not representative sample of the population because of the variety of it. Moreover, describe the contributions of the study more clarify.
  6. At the end of the manuscript, the practical applications/implications of the study (research, policy, and/or practice) should be explained in details.
  7. Finally, 7th Edition APA style need revision because for example doi number are missing and some cited are in the wrong way.

Author Response

Response to the reviews are in bold.

Reviewer 1

The manuscript is a qualitative study that aims to understand the challenges a variety of families living in poverty face when caring for their young children. It is interesting to focus on the understanding of families living in poverty but I have some concerns on the manuscript:

  1. Thank you for your positive comments. We also think this topic is important to include when considering parenting in the 21st

The Current Study section needs revision. It is important that authors mention the previous literature focus on this topic. Please, make sure all objectives and hypotheses are stated in this section and explain the results expected.

  1. We have revised this section and the objectives of the manuscript are more clear.

Create a table would be useful in order to describe the characteristics of the sample.

  1. We have created a table with sample characteristics and we believe this will help the reader.

I believe that Measures section is poorly described and needs to be revision and amplified beyond to carry out an interview guide.

  1. The measures section has been revised and includes more information on the demographics survey as well examples of specific questions that were asked during the focus groups.

Please, revise the limitations of the study, maybe one of these could be that not representative sample of the population because of the variety of it. Moreover, describe the contributions of the study more clarify. At the end of the manuscript, the practical applications/implications of the study (research, policy, and/or practice) should be explained in details.

  1. We have revised the limitation section of the paper and discuss limitations with the sample representation. We also clarify the contributions of this work more directly in the discussion section.

Finally, 7th Edition APA style need revision because for example doi number are missing and some cited are in the wrong way.

  1. The paper has been revised for 7th Edition APA style and all the references have been formatted accordingly.

Reviewer 2 Report

Thank you so much for inviting me to review this manuscript. Authors tried to  understand the challenges a variety of families living in poverty face (i.e., mothers, fathers, teenage parents, mothers in recovery from substance abuse, grandparents raising grandchildren, and Latinx mothers and fathers) when caring for their young children. The manuscript has serious limitations for acceptance. In some parts of the manuscript, the authors use very informal language. Also, there are a large number of sentences that need to be rewritten and revised. The references do not conform to the format of MDPI journals (as far as I know). More description of the sample analysed is needed. There are no tables or figures to show the results (and facilitate the reader's understanding). With such a low sample size, we could apply this to all populations. What about families in other parts of the world? 

Best wishes,

Author Response

Reviewer 2

In some parts of the manuscript, the authors use very informal language. Also, there are a large number of sentences that need to be rewritten and revised.

  1. The informal language has been revised as well as the entire manuscript for clarity and wording.

The references do not conform to the format of MDPI journals (as far as I know). More description of the sample analyzed is needed. There are no tables or figures to show the results (and facilitate the reader's understanding).

  1. As stated above, the references have been revised and the sample and analytical approach has been clarified. We have also added a table regarding the sample characteristics as suggested by reviewers and the Editor.

With such a low sample size, we could apply this to all populations. What about families in other parts of the world?

  1. We have revised the limitations section to discuss generalizability and note that the sample is limited to the U.S. and may not reflect global challenges and concerns.

Round 2

Reviewer 1 Report

The authors were very responsive with regard all my observations but I still have a few comments that I think add value to the manuscript.

  1. Regarding the Current Study section I believe that in the present way is poorly described. I encouraged to authors to make an effort to explain in more detail this important part of the manuscript. For example, it is not clear to me… what are the results expected? ¿What have been found on others studies focused on analyzing the associations between poverty and individual factors of family life or child development?, Do these studies also use qualitative methodology? And, how these studies might connect with the one presented by authors?
  2. Regarging the measures section, again, I believe that still it is poorly described and needs to be revision. It is fantastic that the authors have followed my recommendations, but taking into account that only one instrument is described, which is the “Focus group interview guide”, I think that the authors should explain this section in greater detail and not limit themselves to the extension made at present.

Author Response

Dear Reviewer, 

Thank you for your constructive feedback. After making the revisions suggested, we feel that this paper has increased clarity and better explains the major issues we seek to address.

Responses to the reviews are in bold.

Regarding the Current Study section I believe that in the present way is poorly described. I encouraged to authors to make an effort to explain in more detail this important part of the manuscript. For example, it is not clear to me… what are the results expected? ¿What have been found on others studies focused on analyzing the associations between poverty and individual factors of family life or child development?, Do these studies also use qualitative methodology? And, how these studies might connect with the one presented by authors?

Thank you for this suggestion. We have added additional information to the Current Study section to strengthen our argument for the importance of this research, frame it within the state of existing research, and illustrate how the use of qualitive methods and our diverse sample are a strength to the study. These updates can be found on lines 179-185.

Regarding the measures section, again, I believe that still it is poorly described and needs to be revision. It is fantastic that the authors have followed my recommendations, but taking into account that only one instrument is described, which is the “Focus group interview guide”, I think that the authors should explain this section in greater detail and not limit themselves to the extension made at present.

Thank you for this thoughtful feedback. We have added a detailed description of the interview guide within the Procedure section. The section is titled “Focus group interview guide”, and can be found on lines 225-237.

Reviewer 2 Report

Dear editor,

The authors have not satisfactorily addressed the shortcomings of the manuscript. To me, it should be rejected.

Best wishes,

Author Response

Dear Reviewer, 

Thank you for your consideration.